# 53BP1: Keeping It under Control, Even at a Distance from DNA Damage

**DOI:** 10.3390/genes13122390

**Published:** 2022-12-16

**Authors:** Emilie Rass, Simon Willaume, Pascale Bertrand

**Affiliations:** 1Université Paris Cité, INSERM, CEA, Stabilité Génétique Cellules Souches et Radiations, LREV/iRCM/IBFJ, F-92260 Fontenay-aux-Roses, France; 2Université Paris-Saclay, INSERM, CEA, Stabilité Génétique Cellules Souches et Radiations, LREV/iRCM/IBFJ, F-92260 Fontenay-aux-Roses, France

**Keywords:** 53BP1, homologous recombination, non-homologous end joining, double-strand break repair, lamins, shieldin, PARP inhibitors, BRCA1

## Abstract

Double-strand breaks (DSBs) are toxic lesions that can be generated by exposure to genotoxic agents or during physiological processes, such as during V(D)J recombination. The repair of these DSBs is crucial to prevent genomic instability and to maintain cellular homeostasis. Two main pathways participate in repairing DSBs, namely, non-homologous end joining (NHEJ) and homologous recombination (HR). The P53-binding protein 1 (53BP1) plays a pivotal role in the choice of DSB repair mechanism, promotes checkpoint activation and preserves genome stability upon DSBs. By preventing DSB end resection, 53BP1 promotes NHEJ over HR. Nonetheless, the balance between DSB repair pathways remains crucial, as unscheduled NHEJ or HR events at different phases of the cell cycle may lead to genomic instability. Therefore, the recruitment of 53BP1 to chromatin is tightly regulated and has been widely studied. However, less is known about the mechanism regulating 53BP1 recruitment at a distance from the DNA damage. The present review focuses on the mechanism of 53BP1 recruitment to damage and on recent studies describing novel mechanisms keeping 53BP1 at a distance from DSBs.

## 1. Introduction

Our DNA is continuously exposed to a large variety of threats leading to DNA damage. Among these lesions, double-strand breaks (DSBs) are considered the most harmful. Two main pathways are involved in the repair of DSBs. The first one, homologous recombination (HR), requires a DNA end-resection step and an identical DNA template in the sister chromatid for DNA repair. The second one, canonical non-homologous end joining (C-NHEJ), ligates the broken ends without necessarily using resection and sequence homology. Failure to repair DSBs will result in cell death, senescence, genomic instability, and hence carcinogenesis. A plethora of DNA-damaging agents from exogenous sources, such as ionizing radiation (IR), or endogenous sources can induce accidental DSBs [1]. DSBs can arise during the DNA replication process, during which the replication fork may encounter obstacles that hinder its progression, thus leading to replicative stress [2].

DSBs can also occur physiologically in a programmed manner [3]. Indeed, during meiosis, HR promotes the accurate segregation of homologous chromosomes and increases genetic diversity [4,5]. C-NHEJ is involved during lymphocyte development. T and B lymphocytes experience a process called V(D)J recombination. This mechanism allows the creation of diversity in the antigen receptor genes, by creating DSBs at specific sites, which contain variable (V), diversity (D) or joining (J) coding segments. Mature B cells further diversify their repertoire through class switch recombination (CSR), which also involves C-NHEJ. This process ensures the efficiency of the immune response by modifying the constant region of the antigen receptors without altering the antigenic specificity [6]. In addition, studies have highlighted the possible involvement of physiological DSBs in neurogenesis. In neural stem/progenitor cells, recurrent DSB clusters were identified mainly in long transcribed genes implicated in neural development [7], and in neurons, DSB induction in the promoter of early-responder genes was found to regulate their expression [8].

To prevent genomic instability caused by DNA damage, cells possess a set of mechanisms ensuring the detection, signaling and repair of the lesion, known as the DNA damage response (DDR). The MRE11-RAD50-NBS1 (MRN) complex, the KU70/80 complex and PARP are three sensors for DSBs that all allow further signaling of the DNA lesion [9]. The DDR is regulated by the activity of three phosphatidylinositol-3 kinase (PI3K)-related kinases (PIKKs), i.e., ataxia telangiectasia mutated (ATM) and DNA-dependent protein kinase (DNA-PK), which are, most of the time, activated following DSBs, and ataxia-telangiectasia mutated rad3-related (ATR), which is predominant during replicative stress [10]. The activation of these PIKKs further propagates the signaling of the DSBs and triggers their repair by the two main DSB repair pathways: HR and NHEJ.

Importantly, the choice of DSB repair pathway must be tightly controlled to avoid genetic instability. A key protein of NHEJ, 53BP1 prevents DNA end resection, thus promoting C-NHEJ, and plays a crucial role in the pathway choice (Figure 1). Therefore, 53BP1 recruitment to the DSB is tightly controlled, and sophisticated mechanisms (in general, through chromatin modifications) exist to control 53BP1 in the DSB vicinity. However, little is known about the regulation of 53BP1 recruitment at a distance from DNA damage, including the sequestration of 53BP1 in the nucleus to prevent its access to chromatin. Here, we review the main biological implication of 53BP1 and its downstream effectors, as well as the mechanisms regulating the access and binding of 53BP1 to damaged chromatin. We also focus on the control of 53BP1 at a distance from DSBs that could prevent unscheduled or unappropriated activation of the DDR.

## 2. The Two Main Mechanisms of DSB Repair

As the HR pathway requires the presence of a homologous sequence, it is promoted in the S/G2 phase, during which the sister chromatid can be used as a template. HR occurs via different models all sharing an initial step of single-strand resection at the DSB ends [11]. Unlike HR, NHEJ is active throughout the whole cell cycle and is predominant during G1. C-NHEJ does not require an initial resection step, in contrast to HR and also to the alternative NHEJ (A-NHEJ) pathway. Indeed, A-NHEJ is initiated by single-strand DNA (ssDNA) resection at the DNA ends and depends on the subsequent hybridization of microhomologies distal to the DNA break. Consequently, A-NHEJ is highly mutagenic compared to C-NHEJ [12,13,14,15]. Both HR and C-NHEJ are essential for the faithful repair of DNA DSBs. To ensure proper genome maintenance, the balance between HR and NHEJ is heavily regulated throughout the cell cycle. On one hand, HR must be restricted to the S/G2 phase when the sister chromatid is present to avoid recombination with repeated homologous sequences in G1 [16,17]. On the other hand, uncontrolled C-NHEJ can also generate genetic instability. Indeed, DSB repair via C-NHEJ during the M phase can lead to chromosomal fusion [18].

## 3. 53BP1 Protein

The protein 53BP1 acts in multiple biological processes (Figure 2). It was first identified as a p53-interacting factor [19], but the role of 53BP1 in p53 activity has only recently been elucidated. Indeed, aside from its key role in DSB repair, 53BP1 also participates in the regulation of cell cycle progression. It is essential for normal p53 signaling and contributes to the activation of the G1/S checkpoint dependent on p53 [20]. The 53BP1-p53 pathway requires the deubiquitinase activity of USP28 [20], and it also controls cell cycle arrest following centrosome loss and extended mitosis [21,22,23] (Figure 2A). 53BP1 forms, both in vitro and in vivo, liquid–liquid phase separation (LLPS) condensates [24], an organization of biomolecules, to promote their association and separation from the cellular medium. Interestingly, p53 and USP28 are present in 53BP1 droplets, and preventing 53BP1 LLPS destabilizes p53 and reduces p53 target gene expression [24]. AHNAK, a G1-enriched interactor of 53BP1 [25], restricts 53BP1 chromatin binding and oligomerization and also impedes 53BP1 LLPS condensates. The 53BP1–AHNAK interaction counteracts p53 activity. This interaction depends on ATM and is increased after DNA damage. AHNAK depletion enhances the 53BP1–p53 interaction and p53 activation, leading to apoptosis in cancer cells and to senescence in non-transformed cell lines [25].

In the following sections, we describe other 53BP1 functions with a focus on its key role in the DSB repair choice by preventing DNA end resection. The 53BP1 structure, its main interactors and the mechanisms and regulation of its recruitment to damaged chromatin is also presented.

### 3.1. Implication in Biological Processes

#### 3.1.1. End-Joining Processes: NHEJ, Telomere Fusion, V(D)J and CSR

Even though 53BP1 is not considered a core NHEJ factor as it is not required for all NHEJ-dependent mechanisms, 53BP1 mediates the joining of DSB ends. 53BP1 also mediates the C-NHEJ-dependent fusion of deprotected telomeres [26,27]. By promoting the synapsis of distal ends, 53BP1 controls the movement of deprotected telomeres [26,28] and IR-induced DSBs in conjunction with the LINC complex and dynamic microtubules [29] (Figure 2B). While short or very long distances of V(D)J recombination are not affected by 53BP1 deficiency [28,30], 53BP1 is required for the rejoining of long-range events, with DSBs separated by distances overlapping with γH2AX spreading [30] (Figure 2C). CSR efficiency relies on 53BP1 chromatin recruitment and oligomerization [30,31,32,33,34]. The loop formation between switch regions [35] and the regulatory timing of DSB induction by AID in these regions [36] are also 53BP1-dependent. After DSB induction, 53BP1 mediates C-NHEJ-dependent long-range CSR by protecting ends from resection [37,38,39,40] and prevents A-NHEJ-mediated rejoining of the repetitive intra-switch regions [38] (Figure 2D).

**Figure 2 genes-13-02390-f002:**
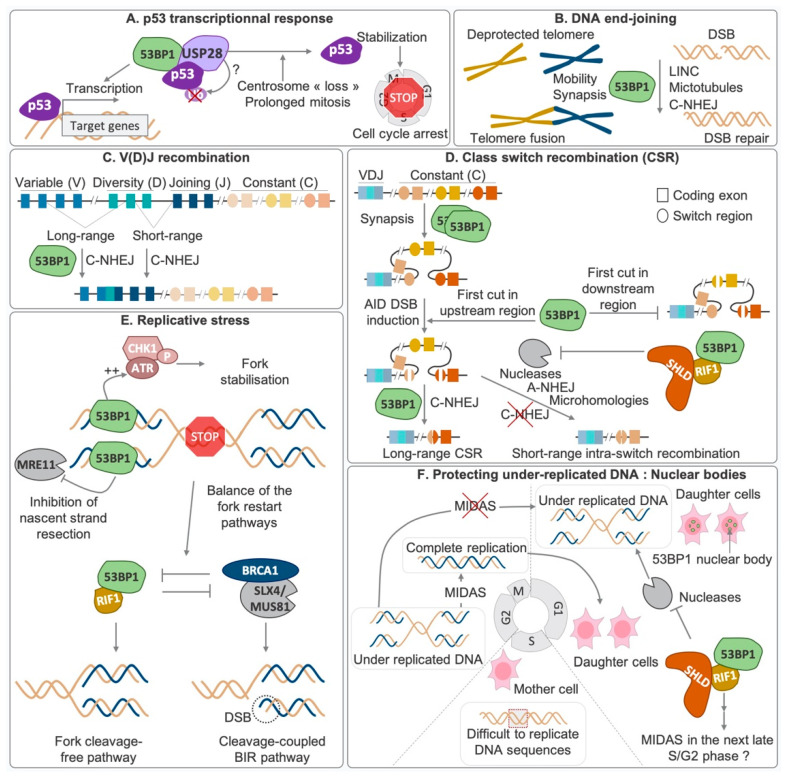
**Implication of 53BP1 in biological processes:** 53BP1 has been implicated in (**A**) the p53 response in different settings; (**B**) C-NHEJ-dependent joining of DSB ends; (**C**) long-range V(D)J recombination; (**D**) multiple steps in CSR: loop formation between switch regions, regulation in the order of DSB induction by AID in switch regions and joining of C-NHEJ-dependent long-range CSR; (**E**) response to replicative stress and stalled fork restart; and (**F**) protecting underreplicated DNA from degradation in 53BP1 nuclear bodies (see text for details).

#### 3.1.2. Inhibition of DNA End Resection as a Control of DSB Repair Choice

Importantly, the choice of DSB repair pathway is pivotal to ensure genome stability and depends on the cell cycle phase, DSBs’ localization, and the nature of the DNA ends. 53BP1 plays a key role in the choice of DSB repair pathway by preventing the early step of resection, at accidental or physiological DSB ends [37,38,41] and at deprotected telomeres [26,33,42]. 53BP1 relies on its partners to either physically protect against resection or counteract nuclease activity, at the initiation or extension step of resection (Figure 3).

Once recruited to DSBs by 53BP1, RIF1 interacts with the phosphatase PP1, which inhibits the initial recruitment of MRN and CtIP and possibly affects MRN nuclease efficiency through CtIP dephosphorylation [43]. Interaction with DYNLL1 stabilizes 53BP1 at DSBs and physically inhibits end resection. DYNLL1 also interacts with the resection machinery (e.g., MRN complex, DNA2, BLM). In vitro analysis also confirms that the DYNLL1–MRE11 interaction inhibits MRE11 nuclease activity [44]. Once resection is engaged, 53BP1 can still prevent its extension through PTIP, RIF1 [30,45,46,47,48,49,50] and shieldin recruitment, which link 53BP1 and ssDNA extremities and block the access to DSB ends [41,51,52,53,54,55,56,57]. Shieldin recruits CTC1–STN1–TEN1 (CST)/POLα/PRIMASE [56] to trigger POLα/PRIMASE-dependent 3′ end fill-in DNA synthesis [58,59,60]. ASTE1, which presents endonuclease activity towards ssDNA and 3′ overhangs, might allow DNA end trimming before POLα/PRIMASE-dependent DNA synthesis [61].

**Figure 3 genes-13-02390-f003:**
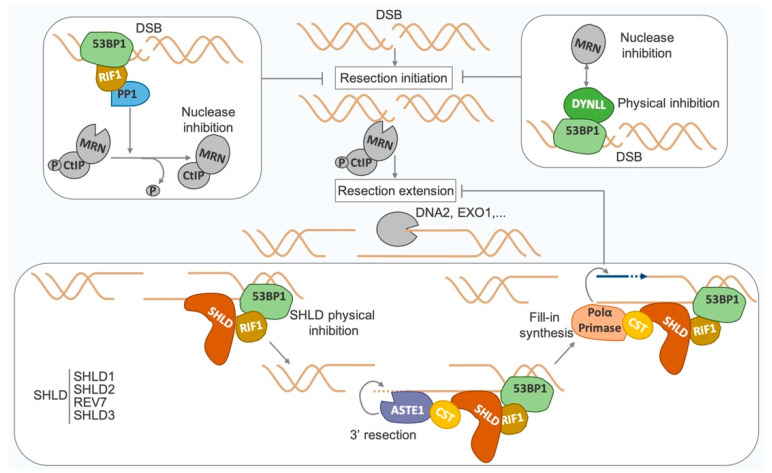
**53BP1 prevents DSB end resection.** 53BP1 and its partners protect against resection. RIF1-PP1 and DYNLL1 prevent the early step of resection (mediated by the MRN complex and CtIP). Shieldin (SHLD) along with CTC1–STN1–TEN1 (CST)/POLα/PRIMASE and ASTE1 protects against the late step of resection (mediated by DNA2, EXO1, etc.) (see text for details).

#### 3.1.3. Replicative Stress and Protection of Reversed Fork

Beside its role in p53 signaling and activation of the G1/S checkpoint dependent on p53 [20], 53BP1 recruitment at stalled replication forks enhances the ATR-CHK1 signaling pathway in response to replicative stress [62]. 53BP1 also protects forks from nascent strand degradation by MRE11 [62,63]. Additionally, 53BP1 competes with BRCA1 for the restart of stalled forks: 53BP1 and RIF1 promote a fast fork restart pathway, which does not require fork cleavage, while BRCA1 facilitates a slower BIR pathway coupled with fork cleavage mediated by SLX4-MUS81 [64] (Figure 2E).

Moreover, DNA sequences that are difficult to replicate, such as chromosome fragile sites (CFS), tend to be replicated during the G2/M phase through mitosis DNA synthesis (MiDAS) [65]. Failure of this mechanism before cell division gives rise to 53BP1 nuclear bodies (NB) in the next G1 [66,67,68,69,70], which protect DNA from degradation through the recruitment of RIF1 and shieldin. RIF1, presumably through its function in replication timing, delays the repair of these regions to the following late S/G2 phase, through MiDAS [71] (Figure 2F).

### 3.2. Structure and Key Interactions

53BP1 is a large protein with no known enzymatic activity, but with several identified domains (Figure 4).

The C-terminal domain is composed of two BRCA-carboxyterminal (BRCT) repeats. This domain mediates the interaction of 53BP1 with p53 [19,73,74] and USP28 [75]. While the BRCT repeats of 53BP1 directly interact with γH2AX [76,77], γH2AX is nevertheless dispensable for the initial 53BP1 recruitment to DSBs [78]. The BRCT repeats seem dispensable for most of the DSB repair activities of 53BP1 [30,79], but are required for the slow phase of DSB repair [80]. These slow kinetics are believed to represent heterochromatin DSB repair and require the ATM-dependent phosphorylation of KAP1, in order to allow chromatin relaxation following KAP1 release [81]. The actual hypothesis is that the 53BP1 BRCT interaction with γH2AX [76,77] and MRN [82] mediates the retention of phosphorylated ATM in the vicinity of heterochromatin DSBs, allowing KAP1 phosphorylation and heterochromatin DSB repair. A role for EXPAND1, which expands the 53BP1 ionizing radiation-induced foci (IRIF) [83] and presents chromatin relaxation abilities, cannot be excluded.

The central domain of 53BP1, called the focus-forming region (FFR), is the minimal domain required for 53BP1 recruitment to DSB sites [84,85] (see Section 3.3). This region is composed of several motifs, all of which are necessary, but not sufficient on their own, to promote 53BP1 accumulation at DSBs: the dynein light chain-binding domain (LC8), the oligomerization domain (OD), the glycine-arginine rich motif (GAR), the tandem Tudor domain (TUDOR), the ubiquitin-dependent recognition motif (UDR) and the nuclear localization sequence (NLS). Interestingly, 53BP1 nuclear import depends on the phosphorylation of a residue within the NLS [86,87], on the nuclear transporter importin β and on the nucleoporin NUP153 [88,89]. The poorly characterized GAR domain is methylated by PRMT1 [90,91]. The OD domain allows the formation of 53BP1 dimers in the nucleoplasm (in the absence of DNA damage) [79,91,92], which can assemble as oligomers at DNA damage sites [93]. The interaction of DYNLL1/LC8 with the LC8 domain of 53BP1 [32,94,95] is also required for the efficient oligomerization of 53BP1 [32]. The TUDOR domain [96] mediates the interaction of 53BP1 with dimethylated lysine 20 of histone H4 (H4K20me2), a constitutive histone mark [97,98,99]. The UDR domain of 53BP1 allows its binding upon DNA damage to the RNF168-dependent ubiquitinated lysine 15 of histone H2A (H2AK15ub) [31].

The N-terminal region of 53BP1 is required for most of its DSB repair functions. Up to 28 S/T-Q sites phosphorylated upon DSBs by ATM and ATR have been identified in this region [30,100,101]. They mediate 53BP1 interaction with its effectors (i.e., PTIP, RIF1). However, the ATM-dependent phosphorylation of these sites seems to be dispensable for 53BP1 foci formation [30,85,102]. Separated groups of phosphoresidues seem to mediate specific interactions. Although the first eight S/T-Q sites seem important for PTIP interaction [45], only serine 25 (Ser25) has been clearly identified as such. The simultaneous mutation of the next seven S/T-Q sites impairs RIF1 interaction [45]. The remaining sites promote DNA ends mobility (MOB domain), although an interacting factor has yet to be identified [29].

### 3.3. 53BP1 Recruitment to Damaged Chromatin

As mentioned earlier, 53BP1 recruitment at DSBs relies on its TUDOR and UDR domains interacting, respectively, with H4K20me2 and H2AK15ub. The TUDOR domain of 53BP1 can interact endogenously with H4K20me2 [97,98]. As this chromatin mark is present under basal conditions in a majority of nucleosomes [103], the lack of H4K20me2 accessibility was assumed to be the reason for the absence of 53BP1 foci without DNA damage. Accordingly, both L3MBTL1 and JMJD2A (also known as KDM4A) mask H4K20me2 in the absence of DNA damage [104,105]. Upon DSB induction, activation of RNF8 and RNF168 induces the ubiquitination of both L3MBTL1 and JMJD2A [104,105]. Ubiquitination of JMJD2A triggers its addressing for proteasomal degradation [105], while ubiquitinated L3MBTL1 is removed from chromatin in a valosin-containing protein (VCP)-dependent way [104,106]. H4K20me2 is thereafter accessible for 53BP1 recognition.

The interaction of 53BP1 with DSBs also depends on RNF168 activity [31]. Indeed, 53BP1, at least as a dimer, recognizes nucleosomes containing both H4K20me2 and H2AK15ub [31]. Residual 53BP1 binding to chromatin is independent of DNA damage [30,97,99,107], H2AX [30] or RNF8 [99], and 53BP1 is initially recruited to DSBs independently of γH2AX [78]. However, its stable IRIF formation requires the chromatin ubiquitination cascade downstream of MDC1 [97,108,109,110]. The initial RNF168-dependent ubiquitination of H2A [111], sustained by RNF8-RFN168 for K63-ubiquitin chain formation [108,109,111], as well as H2AX [78,112], is required for 53BP1 retention at DSBs. 53BP1 spreading around DSBs can reach mega-base distances, with a similar profile to γH2AX and ubiquitin [113].

53BP1 DSB recruitment also relies on its OD domain and on the interaction between the LC8 domain and DYNLL1, which both promote 53BP1 oligomerization [32]. Endogenously, 53BP1 exists as a dimer in the nucleoplasm [93]. After DNA damage, the histone marks H2AK15ub and H2AK20me2, respectively, assist DSB localization of the dimers, and promote their retention. Their combined activity induces oligomerization and foci formation through probable γH2AX stabilization [93]. Super-resolution microscopy analyses further reveal that upon DNA damage, nanodomains of 53BP1 colocalize with topologically associating domains (TAD). RIF1 localization to the boundaries of these nanodomains triggers the formation of circular microdomains around one DSB site [114].

Recruitment of 53BP1 also depends on its posttranslational modifications by different actors. For example, AMPK phosphorylation on serine 1317, localized in the FFR domain, seems to be required for 53BP1 IRIF formation [115]. The involvement of lysine 1268 ubiquitination was also reported. Indeed, one study reported that RFN168-dependent poly-ubiquitination promotes 53BP1 oligomerization before its localization to DSBs [107]. In another study, 53BP1 retention at DSBs appeared to require RAD18-dependent mono-ubiquitination [116].

### 3.4. DSB Recruitment Regulation

#### 3.4.1. Cell Cycle Regulation of 53BP1 Recruitment

The cell cycle phase is also an important factor in 53BP1 recruitment. In mitosis, although the initial signaling of DSBs is correctly executed, with recruitment and activation of ATM, γH2AX or MDC1, there is no recruitment of RNF8, RNF168, 53BP1 or BRCA1 [117]. This can be explained by the inhibition of the MDC1–RNF8 interaction through the CDK1-dependent phosphorylation of RNF8. Interestingly, preventing this RNF8 phosphorylation is sufficient to restore BRCA1 foci formation, but not 53BP1 DSB recruitment [118]. This suggests that a second mechanism prevents 53BP1 recruitment to DSBs in mitosis. Indeed, mass spectrometry analyses have identified, within the UDR domain of 53BP1, two residues that are constitutively phosphorylated in mitotic cells: threonine 1609 (T1609) and serine 1618 (S1618) [118]. S1618 phosphorylation is PLK1-dependent, and T1609 phosphorylation is probably p38 MAPK- or CDK1-dependent [118,119,120]. Interestingly, as cells progress from mitosis to G1, these two sites are dephosphorylated through PP4C/PP4R3β phosphatase [119], whose activation is regulated through its CDK5-dependent phosphorylation [121]. Unphosphorylated T1609 and T1618 are required for the formation of both 53BP1 foci [118,119] and 53BP1 NB [119]. While the inability to dephosphorylate those residues confers IR sensitivity [118,119,121], 53BP1 recruitment to DSBs in mitosis leads to chromosome segregation issues [118,119] and to telomere fusions [118].

Even if the S/G2 phase is a favorable HR environment, 53BP1 can be recruited to DSBs in S/G2 cells, although dilution of the histone mark H4K20me2 on chromatin by replication reduces 53BP1 recruitment to DSBs [122]. As in G1, RIF1 is recruited in S/G2, through the ATM-dependent phosphorylation of 53BP1. Once recruited to DSBs, 53BP1 and RIF1 form a barrier against end resection by MRN and CtIP, thus preventing HR. In G1 or early G2, 53BP1 localizes to the DSB site. In late G2, IRIF are enlarged and 53BP1 is displaced to the IRIF periphery, while the core is occupied by RPA. This repositioning requires the action of BRCA1 and POH1, which relieves, respectively, the barriers formed by 53BP1 and RAP80 [123]. However, 53BP1 phosphorylation and RIF1 recruitment are transient during the S/G2 phase. Indeed, the phosphatase PP4C is recruited in a BRCA1-dependent manner and is responsible for 53BP1 dephosphorylation and RIF1 release during the S/G2 phase [124]. WIP1, another phosphatase, probably works in conjunction with PP4C to promote 53BP1 dephosphorylation following DNA damage in the S phase [125]. Additionally, while in G1 cells, Sp1 is required for 53BP1 IRIF formation [126], upon entry in the S phase, RNF4 sumoylation of Sp1 triggers its proteasomal degradation and 53BP1 removal from DSBs [127].

#### 3.4.2. Regulation of 53BP1 Stability, Recruitment, and Spreading

Several mechanisms are implicated in the control of 53BP1 availability in the nucleoplasm or chromatin as they can trigger 53BP1 proteolysis. Both the protease CTSL from the endosomal/lysosomal pathway [128] and the E2 ubiquitin ligase UbcH7 from the ubiquitin/proteasome pathway [129] control the 53BP1 protein level. UbcH7 mediates 53BP1 degradation in endogenous conditions and following DNA damage, while CTSL is only implicated after DNA damage [130]. More recently, β-arrestin 1 was implicated in 53BP1 degradation, through an interaction with the E3 ubiquitin ligase RAD18 [131]. Following DNA damage, the E3 ubiquitin ligase SPOP binds and polyubiquitinates 53BP1, which is then removed from the chromatin by NPL4, a cofactor of the VCP segregase complex [132]. Interestingly, the DNA damage-dependent PARylation of 53BP1 is recognized and ubiquitinated by the E3 ubiquitin ligase RNF146, which triggers 53BP1 degradation. NUDT16, a member of the Nudix hydrolases, presents hydrolase activity toward PARylated 53BP1 and therefore protects it from degradation [133].

Recruitment of 53BP1 to DSBs is also positively and negatively modulated by different mechanisms, such as other chromatin modifications, binding of enhancer or competitor proteins and downregulation of 53BP1 upstream activators. For example, acetylation of lysine 16 of histone H4 (H4K16) will affect the affinity of the 53BP1 TUDOR domain for the neighboring H4K20me2 [134,135], while TIP60 acetylation of H2AK15 prevents its ubiquitination by RNF168 [136]. On the contrary, after DNA damage, the interaction between the GLP-dependent methylation of H4K16 and the 53BP1 TUDOR domain enhances that of H4K20me2 [137]. Aside from the Tudor-interacting repair regulator (TIRR) (see 4.2), MBTD1, a subunit of the TIP60 complex, competes with 53BP1 for H4K20me2 recognition [136]. On the other hand, 53BP1 binding to H4K20me2 is promoted by an interaction between its TUDOR domain and the kinesin KIF18B [138]. Some ubiquitin ligases also counteract 53BP1 recruitment. Indeed, downregulation of RNF168 through its proteasomal degradation, by the E3 ubiquitin ligases TRIP12 and UBR5, limits 53BP1 spreading on damaged chromatin [139]. The RNF168 paralogue RNF169, whose recruitment to DSBs is RNF8-RNF168-dependent, competes with 53BP1 for the binding to ubiquitinated chromatin [140,141]. Interestingly, the affinity of RNF169 for H2AK15ub is stronger than that of 53BP1 [142]. Many deubiquitinating enzymes can counteract the RNF8-RNF168 signal spreading [143]. Finally, posttranslational modifications of 53BP1 also regulate its recruitment. Indeed, acetylation of the 53BP1 UDR domain inhibits its recognition of nucleosomes [144]. In addition, 53BP1 is sumoylated in a DNA damage-dependent manner, but intriguingly, recruitment of 53BP1 and the SUMO E3 ligase PIAS4 seems to be interdependent [145].

## 4. Control at a Distance from DNA Damage

This section will describe the control of 53BP1 recruitment to damaged chromatin at a distance from DNA damage (Figure 5). Indeed, factors that interact with 53BP1 at a distance from chromatin can sequester 53BP1 in the nucleoplasm. This allows the timely and appropriate recruitment of 53BP1 to chromatin, i.e., when DSBs appear.

### 4.1. FOXK1

Tandem affinity purification analyses have identified FOXK1 as an endogenous interactor of 53BP1 [146]. FOXK1 belongs to the family of forkhead box class K transcription factors and participates in cell metabolism, growth and proliferation. A recent report identified a direct interaction between the OD domain of 53BP1 and FOXK1 [146]. This interaction occurs in the soluble nuclear fraction and is increased following DNA damage in an ATM/ChK2-dependent manner in the S phase. Interestingly, overexpressed forms of FOXK1 that can bind to 53BP1 impair 53BP1 IRIF and reduce the interaction of 53BP1 with its downstream partners RIF1 and PTIP. In BRCA1-deficient cells, these overexpressed forms of FOXK1 recapitulate many of the 53BP1 deletion phenotypes, such as restoration of RAD51 IRIF and partial HR efficiency, but also resistance to PARP inhibitors (PARPi). Thus, the FOXK1–53BP1 interaction varies during the cell cycle and participates in the control of the choice of DSB repair pathway, promoting HR in the S phase and NHEJ in the G1 phase [146].

### 4.2. TIRR

Studies to reveal factors regulating 53BP1 DSB recruitment have identified, through mass spectrometry analyses, TIRR as an interactor of the FFR domain of 53BP1 [147,148]. Although TIRR belongs to the Nudix hydrolase family, a group of proteins that present activity towards a large variety of pyrophosphates, TIRR itself lacks enzymatic activity. As mentioned, the FFR domain of 53BP1 is required for its accumulation at DSBs. TIRR specifically associates with the TUDOR domain, in the FFR region of 53BP1 [147,148]. By interacting with the TUDOR domain, TIRR masks the 53BP1 H4K20me2-binding motif, preventing its accumulation at DSBs and keeping 53BP1 away from chromatin in the absence of DNA damage. This interaction is dissociated after DNA damage. The ATM-dependent phosphorylation of 53BP1 participates in the regulation of the 53BP1–TIRR complex after DNA damage, as the dissociation is prevented by treatment with an ATM inhibitor [147]. Dissociation is also impeded in cells expressing a mutant form of 53BP1 that cannot be phosphorylated on the 28 N-terminal S/T-Q sites. In agreement with 53BP1 defects, loss of TIRR restores PARPi resistance in BRCA1-deficient cells [148]. On the contrary, TIRR overexpression confers resistance to PARPi [147].

TIRR also affects the functions of the 53BP1–p53 complex. Indeed, TIRR regulates the stress-induced interaction of 53BP1 and p53 by competing with p53 for the TUDOR domain of 53BP1, which was shown to interact with dimethylated p53 [149,150]. Loss of TIRR leads to an aberrant increase in the activation of p53 target genes. Additionally, TIRR mRNA levels negatively correlate with the expression of key p53 target genes in breast and prostate cancers. Therefore, TIRR is an important inhibitor of the 53BP1-p53 complex [151].

### 4.3. NuMA

Recently, NuMA has been identified as a regulator of 53BP1 mobility, IRIF formation and function [152]. NuMA is a structural nuclear protein involved in various mitotic activities and acts as a hub in nuclear formation, spindle assembly and maintenance [153]. An interaction between 53BP1 and NuMA, which decreases after DSB induction with IR, has been described [152]. This interaction allows an additional layer of regulation of 53BP1 at a distance from the DNA damage by NuMA. Indeed, 53BP1 mobility is reduced by NuMA in the nucleoplasm and increased in the case of DNA damage. In contrast, NuMA depletion increases 53BP1 mobility. This phenomenon seems to be regulated by ATM, which phosphorylates NuMA on serine 395, likely serving as a release mechanism for 53BP1 [152].

Moreover, it has been proposed that the nuclear insulin-like growth factor 1 receptor (nIGF1R) facilitates a 53BP1-dependent DDR by regulating the NuMA–53BP1 interaction. Indeed, this interaction was reduced in IGF1R-negative mouse embryonic fibroblasts [154].

### 4.4. Lamins

Lamins are type V intermediate filaments and are the main component of a fibrous membrane underneath the nuclear envelope (NE), called the nuclear lamina (NL). Lamins are classified into two types: A-type lamins (lamins A and C) and B-type lamins (lamins B1, B2 and B3). Lamins are present in the NL and, in a smaller proportion, in the nucleoplasm [155,156]. Beside their important role in the structure of the NE, studies have shown that lamins also participate in DSB repair, interact with 53BP1 and regulate its recruitment.

#### 4.4.1. A-Type Lamins

Cells deficient in lamin A/C showed impaired cellular survival in response to DNA damage agents and a persistence of γH2AX foci, thus suggesting a role of lamin A/C in DSB repair pathways [157,158]. LMNA-deficient MEFs showed defective long-range NHEJ [157], which can be explained by decreased protein levels of 53BP1, and also defects in accumulation at DSB sites. These observations were due to a decreased stability of 53BP1 proteins [158]. Indeed, an interaction between lamin A/C and the 53BP1 TUDOR domain has been reported, which promotes the retention of 53BP1 in the nucleus, thus stabilizing 53BP1 and shielding it from UbxH7-dependent proteasomal degradation [130,159]. This strongly suggests that lamin A/C may control NHEJ by regulating 53BP1 levels. Lamin A/C also contributes to 53BP1 stabilization by regulating the levels of the protease cathepsin L. Indeed, there is an increase in the activity and protein levels of cathepsin L in LMNA-deficient MEFs [130]. Furthermore, 53BP1 protein levels are restored upon the depletion of cathepsin L in these cells, thus leading to a restauration of DSB repair [128].

The premature aging disease Hutchinson–Gilford progeria syndrome (HGPS) is caused by an LMNA mutation preventing the conversion of farnesyl-prelamin A to mature lamin A by ZMPSTE24. This gives rise to an immature form of lamin A termed progerin, which accumulates at the nuclear periphery [160,161]. HGPS and ZMPSTE24-deficient cells show defects in DSB repair mechanisms due to defects in the recruitment of DSB repair proteins caused by the presence of progerin or prelamin A. Indeed, HGPS patient cells present a decrease in 53BP1 foci formation [162]. Moreover, NUP153, which participates in 53BP1 nuclear import [88,89], is mislocalized following prelamin A accumulation [163]. The subsequent defective nuclear import of 53BP1 by NUP153 leads to increased cytoplasmic localization of 53BP1, thus preventing 53BP1 recruitment to DSB sites [163].

#### 4.4.2. B-Type Lamins

Lamin B1 also plays a role in 53BP1 recruitment to DSBs and their repair. Depletion of lamin B1 leads to spontaneous 53BP1 and γH2AX foci, suggesting the presence of DSBs. It has been proposed that this could be due to an alteration in the levels of several NHEJ and HR factors. Upon depletion of lamin B1, an increase in 53BP1, BRCA1, ATR, RAD50 and MRE11 protein levels was reported, while there was a decrease in the levels of DNA-PKcs, NBS1 and RAD51 [164]. However, the direct involvement of the misregulation of these factors, specifically upon lamin B1 depletion in DNA damage accumulation, needs to be evaluated. Recently, our team described a new direct interaction between endogenous 53BP1 and lamin B1 [165]. This interaction encompasses the TUDOR and UDR domains of 53BP1 and is dissociated upon DNA damage. Therefore, in the absence of DNA damage, lamin B1 interacts with 53BP1 and acts as a reservoir for 53BP1, keeping 53BP1 away from repair sites but quickly available in the case of genotoxic stresses. Of note, as TIRR, lamin B1 masks 53BP1 chromatin binding domains. Upon DNA damage, 53BP1 dissociates from lamin B1, thus allowing its recruitment to DSBs. This dissociation is dependent on 53BP1 phosphorylation, likely by ATM, since an ATM inhibitor decreases this dissociation, and 53BP1, which cannot be phosphorylated at the 28 S/T-Q sites of the N-terminal domain, is not dissociated from lamin B1 upon DNA damage. However, in the case of lamin B1 overexpression, 53BP1 is sequestered by lamin B1, leading to a defect in 53BP1 recruitment to DSBs. This is associated with DNA damage persistence, chromosome instability, NHEJ repair defects and increased sensitivity to DNA-damaging agents [165]. In contrast, overexpression of a form of lamin B1 that is not able to interact with 53BP1 has no impact on 53BP1 recruitment, DNA damage persistence and chromosome alterations. Interestingly, an altered nuclear shape and increased lamin B1 levels have been reported in several tumor cases and seem to be associated with a poor prognosis [166,167,168,169]. Lamin B1 overexpression causes DSB accumulation, which leads to chromosomal rearrangements and genetic instability [165], both being hallmarks of cancer. This suggests that lamin B1 dysregulation could play a role during the process of tumorigenesis through 53BP1 sequestration and/or DNA damage accumulation.

A recent study also reported that the acetylation of lamin B1 at lysine 134 (K134) impaired the recruitment of 53BP1 to DSBs, thus negatively regulating C-NHEJ. K134 acetylation also induces the persistent activation of the G1/S checkpoint [170].

Together, these studies highlight mechanisms that modulate 53BP1 recruitment at a distance from damaged chromatin after DNA injury, thus precising the link between the nuclear envelope, DSB repair and genome stability.

## 5. 53BP1 Defects in PARP Inhibitor Therapeutic Outcome

HR-deficient cells and tumors, such as BRCA1-deficient cells, are sensitive to PARPi [171,172]. These molecules either prevent the repair of single-strand breaks (SSBs) that are converted by the replication fork into DSBs (which cannot be repaired because of the HR defect), or they trap PARP1 on the SSB, which impedes the progression of the replication fork [173]. Mechanisms allowing HR restoration, such as resection recovery, confer resistance to PARPi. As a consequence, the end protection property of 53BP1 is clearly highlighted in BRCA1-deficient cells, in which a mutation of or deficiency in 53BP1 leads to resistance to PARPi [174].

One can easily understand that mechanisms affecting the correct and timely localization of 53BP1 to DSBs will recapitulate 53BP1 deficiency in terms of resistance to PARPi treatment in BRCA1-deficient cells. Indeed, preventing the dephosphorylation of T1609 and S1618 in the UDR domain, thus impeding 53BP1 DSB recruitment, confers PARPi resistance to BRCA1 mutant cells [119]. PARPi resistance in BRCA1 mutant cells was also observed following deficiencies in 53BP1 downstream effectors, including RIF1 [46,48,49,50], REV7 [53,175], the shieldin complex [53,54,55,56], the CST complex [56,60,176], ATSE1 [61] and DYNLL1 [32,44,95]. However, the formation of the RAD51 filament in BRCA1 53BP1 null cells can be mediated by the RFN168-dependent recruitment of PALB2 and is sufficient to restore resistance to PARPi [177]. Interestingly, preventing the interaction of 53BP1 with PTIP in BRCA1-deficient mice restored resection, while HR efficiency was still abrogated as RAD51 loading was impaired [178]. As a consequence, those cells are sensitive to PARPi. Interestingly, inhibiting the shieldin complex in these cells restores the formation of RAD51 IRIF and leads to resistance to PARPi [178], confirming the previously observed role of the shieldin complex in blocking RAD51 loading [41].

However, the effects of some of these 53BP1 effectors on restoring survival and HR were sometimes less extended than following 53PB1 loss. This might account for the additional properties of 53BP1. Indeed, the MOB domain of 53BP1 is implicated in the mobility of DSBs [29]. Interestingly, following PARPi treatment, the formation of radial chromosomes was decreased in LINC complex-deficient cells depleted of BRCA1, although to a lesser extent than in 53BP1-deficient cells [29].

All these observations suggest that, at least, 53BP1 abilities of promoting DSB mobility and protecting from resection are required to sensitize cells to PARPi in HR-deficient cells. Interestingly, factors controlling 53BP1 recruitment at a distance from damaged chromatin can also trigger PARPi resistance in BRCA1-deficient cells. Indeed, overexpression of TIRR [147] or FOXK1 [146], which both sequester 53BP1, induces PARPi resistance in BRCA1-deficient cells. Moreover, NuMA overexpression, which limits 53BP1 mobility in the nucleoplasm, decreases the formation of chromosome aberrations after PARPi treatment in BRCA1-deficient cells [152].

## 6. Conclusions

The balance between the different DSB repair pathways is important to maintain genomic stability. In light of today’s knowledge, this balance between HR and NHEJ appears as a dichotomy between 53BP1 and DSB end resection. Interestingly, cells have adopted different strategies to allow 53BP1-dependent inhibition of end resection, thus promoting NHEJ. In this function, 53BP1 relies on its interaction with its partners and can prevent resection at both its initiation and extension steps. However, intriguing points remain to be answered about the end-protection role of 53BP1 and its partners, e.g., 53BP1 has been involved in protecting nascent strands from degradation at stalled forks, but whether shieldin or DYNLL1 is implicated is not yet known. RIF1 can also be directly recruited to the SET1A/BODL1-dependent methylation of histone H3 at lysine 4 [179]. Interestingly, BOD1L and SET1A were shown to protect stalled forks from degradation through RAD51 filament stabilization [180,181]. BRCA1 also protects stalled forks from resection by promoting RAD51 binding [182,183]. One can therefore wonder whether, in specific contexts, 53BP1-RIF1 and BRCA1 could actually collaborate to protect stalled forks from degradation until restart mechanisms can take place. Furthermore, the implication of the factors controlling 53BP1 recruitment at a distance from DSBs is yet to be investigated. These factors undoubtedly generate a pool of 53BP1 in the nucleoplasm, which is immediately available when needed. However, evidence is lacking about these factors’ interconnections with each other and also with other partners of 53BP1. Investigating these questions would be of upmost interest to draw a more complete picture of the regulation of 53BP1 recruitment to chromatin.

Over the last few years, diverse classes of RNA have been linked to the DNA damage response (DDR). They have been implicated in various aspects, including the recruitment of DDR actors such as 53BP1 [184,185,186]. Interestingly, a recent study also highlighted that upon DSB induction, TIRR/53BP1 dissociation is dependent on RNA polymerase II and RNA molecules [187]. This study highlights the importance of conducting further investigations on the interaction between the transcriptional/posttranscriptional machineries and 53BP1 and its partners, including those controlling 53BP1 recruitment at a distance from a DSB.

Considering its key roles in the DRR, 53BP1 is unequivocally implicated in cancer. Studies have reported various alterations in 53BP1 protein levels in different cancer specimens. Its expression has also been proposed as a prognostic marker for survival and/or response to various treatments [188]. Somatic mutations of 53BP1 in various cancer types have also been identified in cancer databases [189]. These mutations can lead to truncation or missense mutants of the 53BP1 protein and have been characterized for their role in DDR defects, but their relevance to pathogenicity is yet to be confirmed [189]. Moreover, 53BP1 deficiency was correlated with triple-negative breast cancer status and with BRCA1/2 mutations. A low level of 53BP1 was also correlated with poor survival in breast cancer patients [190]. Interestingly, the deletion of 53BP1 in a BRCA1 mutant model restored tumor cells survival following DNA damage treatments [174,190], especially following PARPi [174]. However, some patients present resistance to this treatment, and various mechanisms have been implicated [191]. In mouse models with BRCA1-deficient tumors, prolonged treatments with PARPi revealed that loss of 53BP1 expression in tumor cells can account for this resistance [192]. Thus, our understanding of 53BP1 regulation highlights new potential therapeutic targets, which could be used in combination with other treatments but also new useful predictive biomarkers for the response of anticancer treatments. In this search for a comprehensive identification, the pathways that regulate 53BP1 recruitment at a distance from DNA damage must be considered, especially as deregulation of those factors‘ expression will sequester 53BP1 in the nucleoplasm, even when DNA damage is present.

In this review, we also discussed the role of lamins, key components of the nuclear envelope integrity, in the control of 53BP1, beside their role in nuclear organization, unraveling another relationship between nuclear integrity and the DNA damage response. Investigations should be carried out to assess the role of the lamins during the course of tumorigenesis or in the outcomes of therapy.

## Figures and Tables

**Figure 1 genes-13-02390-f001:**
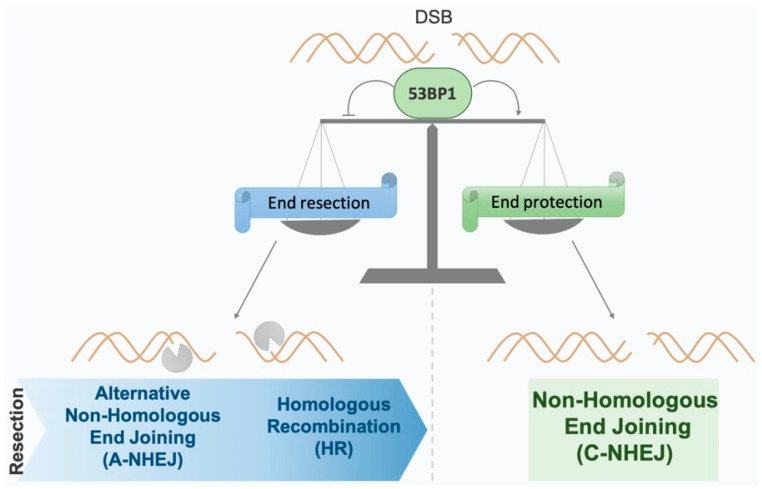
**The importance of 53BP1 in the choice of double-strand break (DSB) repair pathway**. 53BP1 plays a pivotal role in the balance of the DSB repair choice by limiting DSB end resection, thus promoting canonical non-homologous end joining (C-NHEJ) over homologous recombination (HR) and the mutagenic alternative NHEJ (A-NHEJ), which both require a first resection step.

**Figure 4 genes-13-02390-f004:**
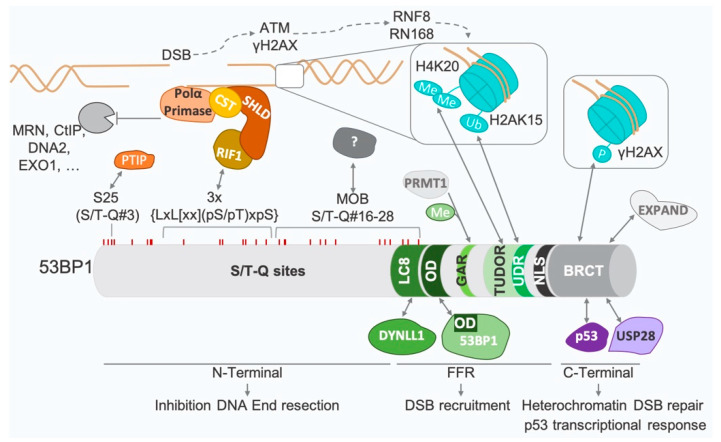
**Domains and interactions of 53BP1.** Upon DSBs, activation of ATM, unmasking of dimethylated lysine 20 of histone H4 (H4K20me2) and ubiquitination of histone H2A on lysine 15 of (H2AK15ub) by RNF168 and RNF8 allow the recruitment of 53BP1 to damaged chromatin. 53BP1 recruitment requires the focus-forming region (FFR), which includes the oligomerization domain (OD), the glycine-arginine rich motif (GAR), the tandem Tudor domain (TUDOR), the ubiquitin-dependent recognition motif (UDR), the dynein light chain-binding domain (LC8) and the nuclear localization sequence (NLS). DSB end resection is inhibited by RIF1, shieldin, CST, POLα and PRIMASE, which interact with the S/T-Q sites on the N-terminal part of 53BP1. RIF1 recognizes three consensus sequences, each containing two leucine and two S/T-Q sites (3x {LxL[xx](pS/pT)xpS}) [72]. Serine 25 (S25) allows the interaction of 53BP1with PTIP. The mobility domain (MOB) also encompasses S/T-Q sites and regulates DNA end mobility, but its interactors remain to be identified. The C-terminal part of 53BP1 contains two BRCA-carboxyterminal (BRCT) repeats, allowing 53BP1 interaction with p53, γH2AX, EXPAND and USP28.

**Figure 5 genes-13-02390-f005:**
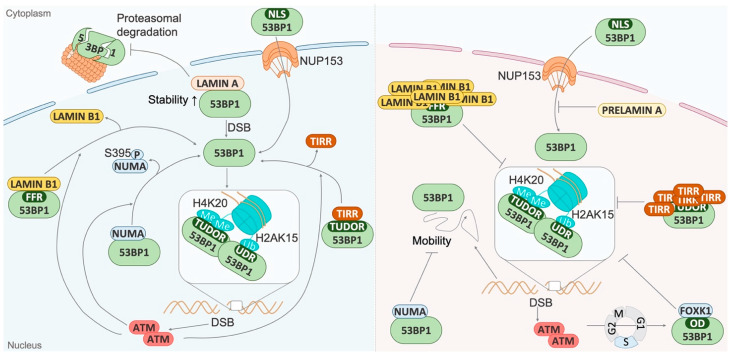
**Factors participating in the regulation of 53BP1 recruitment to the chromatin, at a distance from the DSB.** Left panel: In the absence of DSBs, several factors such as TIRR and NuMA interact with 53BP1 to prevent its recruitment to chromatin by binding region essentials for 53BP1 recruitment. Upon DSBs, mechanisms regulated by ATM take place to dissociate these factors from 53BP1 and allow its recruitment to the damaged chromatin. Nuclear envelope proteins also play an important role in genome stability and 53BP1 regulation. Lamin A interacts with 53BP1 and increases its stability by preventing 53BP1 degradation by the proteasome. Furthermore, 53BP1 interacts with lamin B1 via its FFR domain, impeding its recruitment to DSBs. Upon DNA damage, an ATM-dependent dissociation occurs, thus enabling 53BP1 to be recruited to damaged chromatin. Right panel: In the absence of DNA damage, NuMA associates with 53BP1, reducing its mobility and its access to chromatin. Following DNA damage, ATM enhances the FOXK1–53BP1 interaction in the S phase, preventing 53BP1 recruitment to DSBs. On the other hand, prelamin A expression impedes 53BP1 importation to the nucleus via NUP153. 53BP1 interactors’ deregulation, such as TIRR or lamin B1 overexpression, also inhibits 53BP1 recruitment to damaged chromatin through its sequestration.

## Data Availability

Not applicable.

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
