# Peer review of "53BP1: Keeping It under Control, Even at a Distance from DNA Damage"

_genes, 2022, doi:10.3390/genes13122390_

Round 1

Reviewer 1 Report

Rass et al, comments 2022

This review by Rass et al, focuses on mechanisms that regulate recruitment of 53BP1 and its interactions with damaged chromatin. Simple recruitment of 53BP1 is essential for DNA damage repair pathway choice and is very widely studied. In addition, DNA damage mechanisms are extensively reviewed elsewhere. The unique and intriguing angle of this review, which sets it apart, is the focus on mechanisms that regulate 53BP1 “at a distance from DNA”. At the very outset it is unclear what that entails, but presumably this would be either sequestration of 53BP1 from undamaged DNA and/or interactions with chromatin away from the damage site. Since this is the focus of the review as evidenced in the abstract, it should be clearly enumerated and explained. Otherwise, the initial few sections are generally reviewing DNA damage mechanisms and 53BP1 functions, which is written knowledgeably and exhaustively, but it takes a while to get to the point of the new material that this review is focused on (Section 4). This may cause readers to be fatigued prior to reaching the subject focus. I recommend that the authors explain the recruitment pathways, especially which pathway is recruiting 53BP1 to what and define “at a distance”. In addition, authors should summarize and compact details in the first few sections that have been known a while and needs to be less dense and more conceptual. To be accessible to the generally broad readership of a journal such as Genes, this review needs much re-structuring and simplification as well as more focus on the main subject matter. Some major comments are below:

Major comments:

1. It is unclear what the authors mean by at a distance from DNA damage? Recruitment to the damage site from the cytosol? Or sequestering it away from DNA to avoid damage? They allude to it a bit in Lines 69-72 but it is very vague. This is only addressed first in section 4 but it should be clarified in the intro since the abstract and title specifically says the focus is recruitment and recent pathways for keeping 53BP1 at a distance from DNA damage. 

2. Overall, details in the second section can be compacted and simplified unless the authors require those details for understanding 53BP1 regulation? These pathways are very well known and can be conceptually presented as opposed to listing all the molecular players. 

3. Line 109-123- The opening sentence of these two paragraphs states that the balance between HR and NHEJ must be tightly regulated to ensure proper genome maintenance. The authors then go on to provide instances where bothHR and NHEJ are inaccurate means of repair or cause genome instability in some context. So, it is perplexing how a balance between two error prone mechanisms maintains genome integrity? Because, following the author’s logic, each process has some number of inherent errors, so how does a balancing act ameliorate those errors?? This is a very important point because the authors introduce this balance as being the motivation for investigating 53BP1, but it is contradictory to this paragraph. This must be clarified.

4. Section 3.2 can be shortened and restructured to just focus on key interactions that affect its recruitment and downstream pathway choice. This will help readers keep focused on the review topic. The whole implication in biological processes seem a bit drawn out given that most of this should be about recruitment and sequestration of 53BP1. I am sure telomere fusions and CSR is an important implication but these are well known in the field already. For e.g. the most interesting part of this section was the fact that VDJ dependence on 53BP1 is distance dependent. That seems central to the theme in the review abstract and less discussed as the authors rightly point out. It perhaps deserves more discussion about why the authors think this is true? Especially as CSR seems to be long range dependent. Presenting this part as a comparison of short range vs. long range VDF and CSR dependencies on 53Bp1 would elevate this review.

5. I think the LLPS section is a timely addition given that that topic is spoken about a lot. But could the authors clarify how much of the evidence is in vitro versus in vivo and what parts are deduced by optogenetic manipulations and artificial recruitments? It is a good idea to present caveats to LLPS studies, if any, at the very outset, as should be done for any section.

6. The role of cell cycle in 53BP1 recruitment is interesting. Perhaps the authors can clarify if anything is known about the kinetics of BP1 vs BRCA1 recruitment? Is one faster than the other and does that depend on the cell cycle? Are both proteins constitutively expressed or restricted to certain parts of the cell cycle? BRCA1 aids in the release of 53BP1 from DNA leading to HR as the major pathway in S/G2 Does this imply that 53BP1 is less capable of inhibiting end resection at this stage of the cell cycle? All of these answers would enhance the section even if they are unknowns. Especially because this is what the review abstract says the focus is: recruitment of 53BP1 to breaks.

Some of this is discussed later in the replication stress section but it is still confusing. For e.g. the authors mention in line 293 that 53BP1 favors resection and HR-mediated fork restart, but in line 300 they say 53BP1 competes with BRCA1 for fork restart- through inhibiting resection, I am guessing? Seems contradictory and probably has some sort of situational resolution in context, but it needs clarification. Perhaps a better approach would be to restructure these sections as cell cycle dependent recruitment of 53BP1 and then describe how it’s function can change over the cell cycle and affect pathway choice in each stage of the cell cycle.

It feels like the main focus of the review is section 3.3 and 4. This is where recruitment is discussed. All previous sections were mostly informational, detail heavy without much speculation or future direction or concept, which may make it harder to keep engaged with the central story. I suggest that in the previous sections, the details be simplified conceptually and the parts that are relevant to recruitment and pathway choice be enhanced in terms of cell cycle stage and context of repair. Like lines 493-503 seem reminiscent of the cell cycle control section and could be consolidated in one section for cell cycle control to remind readers of the total picture. 

7. Section 3.3.3: By stability do the authors mean turnover of 53BP1 at the break or the cytoplasmic oligomers or both? Is increased stability just indicative of reduced proteolysis?

8. Line 541: What do the authors mean when they say “control the regulation of recruitment to damaged chromatin at distance to DNA damage?” The section mostly deals with factors that sequester 53BP1 so that inappropriate recruitment of 53BP1 to undamaged DNA does not occur. This should be made very clear. I believe what the authors are actually reviewing is how the soluble fraction of 53BP1 is sequestered away from DNA until breaks are formed and recruitment is necessary. This must be stated clearly. 

9. The conclusion can definitely be enhanced. What are the future directions that the field should go towards?  Can the authors speculate how the current knowledge informs how we think about DNA damage repair choice and the balance for genome integrity?

Minor comments:

Although this is a minor point, the flow and understanding of the article would be better if the English grammar were checked thoroughly, since very often it can change the meaning of a sentence. I pointed out a few consistent issues below but extensive editing would help with this and greatly increase clarity. 

1. Title should be fixed, missing an article: Keeping it under control even at a distance from DNA…

2. Line 12: “physiological processes…”

3. Line 17: suggested change in sentence “53BP1 promotes NHEJ over HR…” Replace favorizing with promoting everywhere.

4. Line 55: “Signaling and repair…”. In general, I would replace signalization with signaling (verb).

5. Line 199: “repair pathway is pivotal…”

6. Line 451: Please expand VCP

7. Line 526: “Competes with 53BP1 for H4K20me2…”

There are many more that are not listed here. 

Reviewer 2 Report

The review titled "53BP1: keeping it under control even at distance from DNA damage" addresses a little-addressed issue with 53BP1.

Comments that authors must attend to are listed below:

1) It is recommended that the authors consider that the topics developed in sections 1, 2 and 3 are consistent with the abstract and title of the review. However, there is a lack of clarity in sub-sections 3.2.1, 3.2.3, 3.2.4, 3.2.5, 3.2.6 and 3.2.7. It is suggested that they include images to give clarity.

2) They need to rearrange sections 3 numeration, because sub-section 3.2.2 is missing.

3) It is important that the authors take in account, that this information is presented without specify if correspond to a physiological context, or cell type, organismic, etc.

4) "Section 5. 53BP1 in cancer and consequence of the 53BP1 defect in the therapeutic result against cancer" must be developed differently to be consistent with the title and abstract of the review. Alternatively, authors can omit it.

5) The authors must re-write the conclusions, since their content does not reflect what was addressed as central issues of the review and lacks consistency with the abstract and title.

6) It is confusing that the topic of section 5 is presented with details alluding to cell types and models, when the information in the previous sections does not.

7) The authors should improve the conclusion, perhaps they could also include a section that comments on the alternatives in transcriptional and post-transcriptional regulation processes that may participate in the remote functioning of the DNA damage site by 53BP1.

Round 2

Reviewer 1 Report

The authors have satisfactorily addressed most but not all comments. However, given that there is a good concept behind the review, I have no further comments.